# Contamination detection in genomic data: more is not enough

Luc Cornet[1] and Denis Baurain[2*]

*Correspondence:
denis.baurain@uliege.be
[2] InBioS–PhytoSYSTEMS,
Eukaryotic Phylogenomics,
University of Liège, Liège,
Belgium
Full list of author information
is available at the end of the

## Abstract

The decreasing cost of sequencing and concomitant augmentation of publicly available genomes have created an acute need for automated software to assess genomic contamination. During the last 6 years, 18 programs have been published, each with its own strengths and weaknesses. Deciding which tools to use becomes more and more difficult without an understanding of the underlying algorithms. We review these programs, benchmarking six of them, and present their main operating principles. This article is intended to guide researchers in the selection of appropriate tools for specific applications. Finally, we present future challenges in the developing field of contamination detection.

**Keywords:** Contamination detection, Genomics, Databases, Algorithms, Review, Corroboration

## Introduction

During the last decade, the number of publicly available prokaryotic genomes has increased dramatically, roughly doubling each year [1]. While this deluge of data has opened new research perspectives in comparative genomics and related fields, it has been accompanied by the growing issue of the contamination of a number of genomes released in public databases [1–4].

The genome of a single organism is supposed to contain only genomic sequences from this organism, and the inclusion of foreign sequences along these genuine sequences is termed "genome contamination". Mis-affiliation of individual sequences can be at the origin of various biases and false inferences. One of the most famous cases is the artifactual report of an important rate of horizontal gene transfer (HGT) in the tardigrade genome [5], which was actually due to overlooked bacterial contamination [6–8]. Contamination has also been reported in genomes of model organisms used by a large community, such as *Nematostella vectensis* [9] and Drosophila [10]. Not only bacterial contamination occurs, but human DNA has also for instance been detected numerous times in non-human databases [11–13]. The presence of foreign DNA in metagenomes

is an important problem for microbiome studies [14]. Genomic contamination is also known to be a source of artefacts in genome skimming [15] or in phylogenomic studies, with emblematic examples of incorrect results in high-profile articles about animal [16, 17] and plant evolution [18, 19]. Moreover, contaminated sequences have the power to spread into and across databases over time [2, 12].

The introduction of foreign sequences can occur at many different steps of the sequencing process, from organism culture to data processing (see Fig. 1 for more discussion). The contamination of an axenic *culture* by unwanted organism(s) and the inclusion of unwanted DNA either during *DNA extraction* or *sequencing* on shared platforms are well-known causes of genomic contamination [2, 20, 21]. Yet, less obvious sources of contamination do exist, like the sequencing of *chimeric organisms* [22] or the presence of plain *taxonomic errors* in reference databases [3]. The contamination can also appear after the sequencing per se, notably during the *in silico* processing of the data. The risk of in-silico contamination is higher when the data comes from metagenomic analyses where the DNA of multiple organisms is extracted in bulk. Indeed, such data can lead to chimeric sequences by merging similar genomic regions during *metagenomic assembly* [1, 23, 24]. The *metagenomic binning* (i.e. the partition of sequences from the constitutive organisms into individual Metagenome-Assembled Genomes (MAGs)) also results in some degree of contamination by lumping in a single MAG contigs reconstructed from different organisms [23] (Fig. 1). All these sources of contamination can be summarized

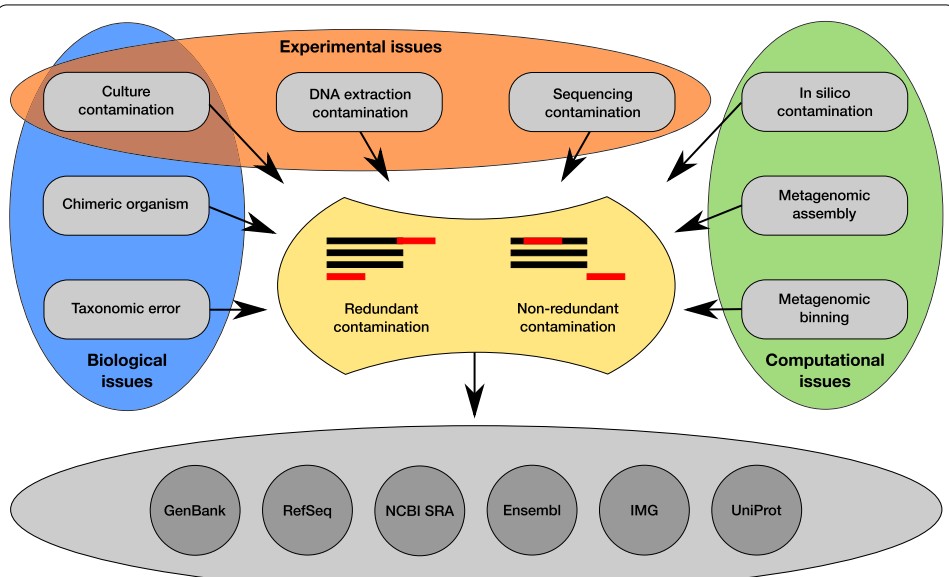

**Fig. 1** Sources of genomic contamination. Three types of issues lead to contamination of genomic sequence data: *biological*, *experimental* and *computational*. The contamination of "pure" cultures can be due to both experimental (e.g. accidental introduction of contaminating microorganisms) and biological causes (e.g. the presence of an endosymbiont). *Redundant* contamination occurs when a genomic segment is present multiple times in a genome (e.g. multiple SSU rRNAs from different organisms). *Non-redundant* contamination occurs when a genomic region of the main organism, the expected one, is replaced by the corresponding region of a foreign organism (e.g. the SSU rRNA of the main organism is replaced by the SSU rRNA from a foreign organism). An extra DNA segment, not part of the main organism but belonging to a contaminant, would also be considered as a non-redundant contamination (e.g. eukaryotic DNA in a bacterial genome). A mixed scenario is also possible, as represented in the redundant contamination part of the figure

in two main types of contamination at the genomic level: redundant and non-redundant [1]. Redundant contamination occurs when a genomic segment is present multiple times in a genome assembly, due to the inclusion of homologous genomic regions from foreign organism(s). In contrast, non-redundant contamination occurs when an extra genomic segment is present in the assembly. Two sub-cases can then be distinguished: (1) a genuine genomic segment is lacking in the target organism (i.e. the completeness is not optimal) and is replaced by a foreign genomic region harbouring (some of) the expected genes or (2) an extra genomic region, for which no homologous region exists in the target organism, is present due to the inclusion of a taxonomically distinct organism (e.g. genomic regions from another kingdom) (Fig. 1).

During the last 6 years, no less than 18 algorithms for the detection of genomic contamination have been published. A majority (11) of these tools have been published or updated during the last 3 years, which complicates determining which programs are currently optimal. Three recent studies, of which two from the first half of 2021, recommend the use of multiple tools to achieve accurate detection [1, 3, 22]. In this review, we first summarize the basics of each tool as well as their specific advantages, so that researchers can make an informed choice when trying to deal with genome contamination. Then, we paint a general overview of the important concepts and open challenges of the field.

## Overview of algorithms

The algorithms can be divided into two main categories, depending on if they are database-free or, in opposition, if they rely on a reference database. The second category contains two different types of tools: genome-wide approaches and estimators based on single-copy gene markers (Fig. 2). All the programs reviewed in this article estimate the contamination level after genome assembly, at the exception of three software packages (*Kraken*, *CLARK*, *CONSULT*), which can also perform read filtering before genome assembly. The majority of the tools described below work on prokaryotes, while *EukCC* only works on Eukaryotes. Tools able to perform inter-domain detection (i.e. to simultaneously deal with prokaryotic and eukaryotic sequences) are indicated in Fig. 2. Binning algorithms commonly used in metagenomics to produce MAGs are not considered in this review because they are not designed to return individual genome contamination statistics.

### Database-free methods

The four programs (*BlobTools*, *Anvi'o*, *ProDeGe*, *PhylOligo*) of this category partition sequences according to the inherent nature of DNA, even if the majority (all at the exception of *PhylOligo*) also rely on taxonomy, labelling sequences for visualization or program calibration, to help with the partitioning. All the programs of this section require a case-by-case inspection by the user and are thus difficult to use for large-scale projects (Fig. 2). Database-free tools can detect both redundant and non-redundant contaminations.

*BlobTools*, initially published in 2013 under the name *Blobology* [25], permits the visualization of sequences from low complexity metagenomic assemblies [26]. The program relies on Guanosine+Cytosine (GC) content and read coverage to represent contigs on

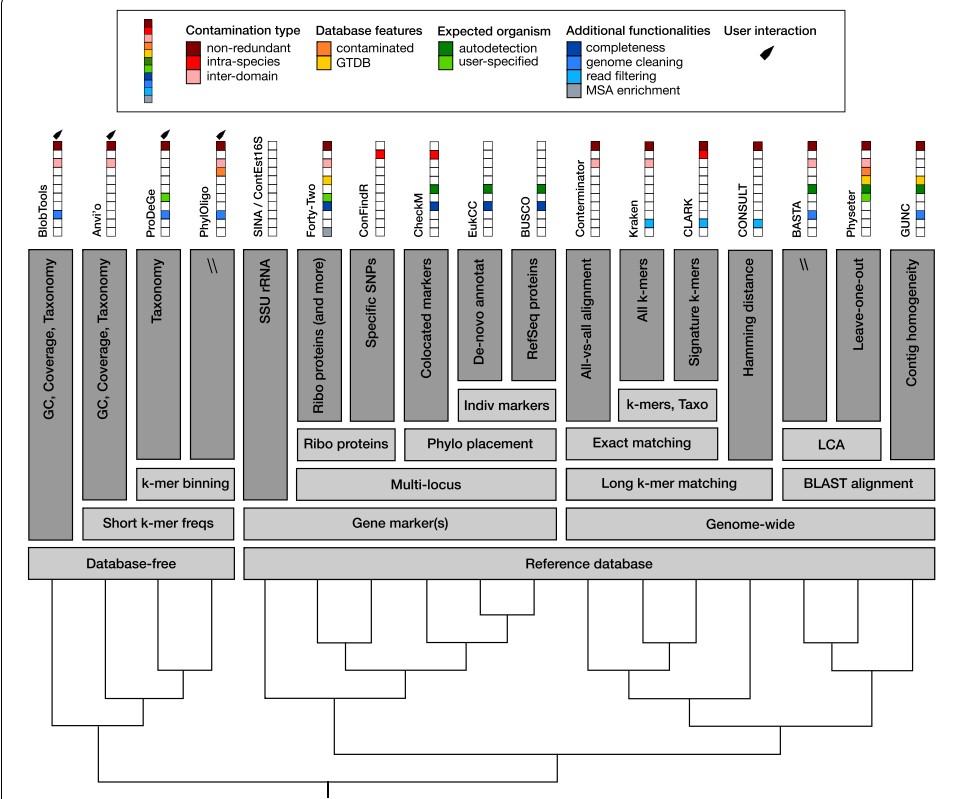

**Fig. 2** Overview of algorithms. The algorithms are clusterized based on their operating principles, as described in the section "Overview of algorithms". Squares on the top of the figure represent specific features of the algorithms. *Non-redundant* means that the software can detect contaminant genes without equivalent in the surveyed genome. *Intra-species* means that the algorithm can detect contamination at the species level. *Inter-domain* means that the algorithm can detect prokaryotic and eukaryotic contamination simultaneously. *Database features* show that the algorithm can use the GTDB Taxonomy and/or a moderately contaminated reference database. *Expected organism* indicates whether the algorithm can detect the main organism by itself and/or if the user can specify it. *Additional functionalities* list interesting peculiar functions of the programs, such as outputting the completeness of a genome, cleaning a genome from its contaminants, filtering reads based on their taxonomy (positive filtering), or enriching Multiple Sequence Alignments (MSAs) in orthologous sequences while controlling the taxonomy

a bi-dimensional plot. Sequences in this plot are coloured according to the NCBI Taxonomy, which has to be provided by the user [26]. The authors state that sequence taxonomy can be obtained using tools such as *BLAST* [27] or *DIAMOND blast* [28]. In that sense, *BlobTools* is not completely independent of a reference database, even if the latter is not integrated in its workflow (Fig. 2). *BlobTools,* which works on prokaryotic genomes, has recently been upgraded, under the name *BlobToolKit,* to support eukaryotic genomes too [29].

The other three programs (*Anvi'o, ProDeGe, PhylOligo*) use short k-mers (i.e. DNA words of length 4 to 9 nt) frequencies to separate sequences. First, *Anvi'o* is a well-established visualization tool in the metagenomic field [30]. It allows the user to represent sequences in an interactive "circoplot" and to organize them according to various sets of parameters [30]. It uses a combination of k-mer frequencies (4-nt long) and read coverage to cluster the contigs, which help identify contaminants [7]. *Anvi'o* also provides

a workflow to perform genome decontamination (see https://merenlab.org/2015/06/25/screening-cultivars/). As for *BlobTools*, a NCBI Taxonomy layer, which in this case is built in *Anvi'o*, can be added by the user to the visualization (Fig. 2). Although this program works on prokaryotic genomes by default, it can handle eukaryotic genomes, but the user has to carry out protein annotation separately.

The last two tools of this section (*ProDeGe*, *PhylOligo*) are conceptually very similar, since they both only rely on k-mer-based binning of sequences (k-mer size is user-specified). The partitioning is computed after a step of calibration of the k-mer frequencies using a sample of sequences from the target organism. *ProDeGe* calibrates the k-mer frequency profile by considering the taxonomy of the expected organism, which has to be provided by the user [31]. Taxonomy is assigned to each contig by homology search (*BLAST*) against a curated version of the Integrated Microbial Genome (IMG) database [32]. If the taxonomy corresponds to the label provided by the user, the contig is considered uncontaminated and serves to calibrate the k-mer frequency profile, which in turn will generate the binning [31]. In contrast, *PhylOligo* does not use a taxonomy to build its profile [33]. Instead, it provides a Neighbour-Joining tree, in which leaves correspond to sequences, and lets the user select contigs that will serve for calibration [33]. To this end, the k-mer frequency profile of each contig is computed and a pairwise distance matrix is used to build the tree [33]. The user can then select the main organism in the tree, with the assumption that it corresponds to the target organism, and contigs are automatically loaded to calibrate the k-mer frequency profile and generate the binning [33] (Fig. 2). *ProDeGe*'s taxonomy is restricted to prokaryotes while *PhylOligo* can work on eukaryotic genomes too.

### Methods associated to a reference database

#### *Gene marker-based estimators*

The seven programs (*SINA*, *ContEst16S*, *Forty-Two*, *ConFindR*, *CheckM*, *EukCC*, *BUSCO*) of this section rely on widely distributed gene markers to assess redundant contamination and non-redundant contamination for *Forty-Two*. These genes are present in a single copy in nearly all organisms and the presence of multiple copies is thus indicative of such type of contamination (Fig. 2).

The simplest approach is to assess the number (and congruent taxonomy) of SSU rRNA genes present in a genome assembly with *SINA* [34]. This has been done notably to estimate the level of the contaminants in cyanobacterial genomes, in corroboration with other methods [22]. The use of this single locus is not frequent because it entails a higher risk of missing contaminants. *SINA* can work on prokaryotic and eukaryotic genomes, albeit separately. The same strategy is also used by *ContEst16S*, available as a website and restricted to prokaryotes [35].

Building upon the same idea, but extended to ribosomal protein reference databases, *Forty-Two* [36] uses *BLASTP/BLASTX* searches to roughly estimate contamination and completeness levels in genomes and transcriptomes. Taxonomic affiliation is based on a *MEGAN*-like algorithm [37] that infers a last common ancestor (LCA) from the set of reference sequences best matching each contig or transcript of the evaluated dataset. *Forty-Two* can be used on prokaryotic or eukaryotic datasets, depending on the reference database considered, RiboDB [38, 39] or a set of manually curated eukaryotic

alignments [40], respectively. Recently, an inter-domain dataset has been assembled. The user can choose between a purely descriptive mode or a mode actively looking for contaminating sequences based on an expected organism. *Forty-Two* further supports the Genome Taxonomy Database (GTDB) [41], in addition to the NCBI Taxonomy. Finally, it is worth mentioning that the initial purpose of *Forty-Two* is orthologous enrichment of Multiple Sequence Alignments (MSAs) for phylogenomic applications [36, 42].

*ConFindR* identifies contaminants by using variations in 53 ribosomal proteins that are used in ribosomal multilocus sequence typing (rMLST) [43]. *ConFindR* first checks the presence of multiple genera in a sample by comparing raw reads, using the *Mash* "screen" option [44], on a custom version of NCBI RefSeq [45, 46] reduced to one genome per bacterial species [43]. *ConFindR* reports cross-genus contamination and does not process further the data if more than one genus is present [43]. Otherwise, intra-species contamination is then estimated by extracting rMLST data from raw reads and assessing the presence of multiple alleles by SNP calling [43]. Among all tools, *ConFindR* presents the highest sensitivity to detect intra-species contamination, which have been reported to be the most damaging source of contamination for clustering analyses, such as phylogenetics or single-nucleotide polymorphism (SNP) discovery [47] (Fig. 2). The program can work on prokaryotic genomes only.

The last three programs of the current category (*CheckM*, *EukCC*, *BUSCO*) use phylogenetic placement to select lineage-specific sets of gene markers. The advantage of this approach is that more genes can be used when the taxonomy is more precise. In practice, genomes are placed on nodes of a precomputed phylogenetic tree to select the most appropriate markers. As an expected number of these makers should be present in a genome assembly, these three programs are able to estimate the completeness of the genomes [48–50]. The first tool to have implemented this strategy is *CheckM* [48], by far the most cited software package of this review. *CheckM* also differs from the two other tools by using colocated sets of markers. Reportedly, markers that are spatially close give a more robust estimation compared to isolated markers [48]. Gene markers and the phylogenetic tree have been computed once for all from a curated version of IMG [32]. *CheckM* begins by extracting the ribosomal proteins to determine the phylogenetic position of the genome under study in its reference tree. Contamination and completeness are then estimated using specific markers based on this placement [48]. It reports intra-species contamination through the "strain heterogeneity" value, which increases when the amino acid identity between two redundant markers is high [48] (Fig. 2). *CheckM* only works on prokaryotic genomes.

The last two programs (*EukCC*, *BUSCO*) use individual gene markers and are similar in terms of methodology. *EukCC* is designed to estimate contamination and completeness in eukaryotic genomes [49]. It uses fungal and protist genomes from NCBI RefSeq [45, 46]. *EukCC* mimics the functionalities of *CheckM* and was the first tool to perform phylogenetic placement for eukaryotes, based on 55 (undocumented) single-locus marker genes [49]. Nevertheless, since version 5, *BUSCO* is also able to perform such kind of placement, but using an unreported number of single-gene markers [50]. For eukaryotic genomes, the main difference between the two programs is the use of RefSeq proteins in *BUSCO* [50] while *EukCC* used *GeneMark-ES* [51] for de novo protein annotation during database construction, which is supposed to improve the accuracy of the estimation

[49]. Another difference between the two tools is that *BUSCO* can also work on bacterial genomes, separately from eukaryotes [50], whereas *EukCC* is devoted to the latter domain (Fig. 2).

### Genome-wide approaches

The seven programs of this section use the entire genome to perform alignment against a reference database. The alignment step can use k-mer matching (*Conterminator*, *Kraken*, *CLARK*, *CONSULT*) or involve longer genomic regions and *BLAST*-like algorithms (*BASTA*, *Physeter*, *GUNC*) (Fig. 2).

The programs based on k-mer matching (*Conterminator*, *Kraken*, *CLARK*, *CONSULT*) use longer k-mers than those of database-free methods, the minimal length being 21 nt. At the exception of *Conterminator*, these tools are the only ones able to filter reads based on taxonomy and thus to estimate the contamination level before, but also after, genome assembly (Fig. 2). At the exception of *CONSULT*, their methods require exact k-mer matching. *Conterminator* is a tool designed to detect cross-domain ("across kingdoms" according to the wording of the authors) contamination in sequence databases by an all-vs-all alignment [2]. Sequences have to be taxonomically labelled for the program to work. *Conterminator* can process large databases thanks to the use of *Linclust* [52], which minimizes the comparison time by first grouping sequence segments if they share canonical (i.e. independent of the strand) k-mers. Ungapped alignment of representative sequences is then performed with *MMseq2* [53] and contaminants are detected based on a minimal identity threshold [2]. Although it has been only tested in cross-domain detection, *Conterminator* can in principle work at any taxonomic level, whether on nucleotide or protein sequences (Fig. 2).

The other two programs using k-mer matching (*Kraken, CLARK*) are not strictly speaking designed for contamination detection. Their initial purpose was to classify reads in metagenomic studies. Nevertheless, their unique architecture makes them suitable for detecting contaminants [12, 22]. The first of these programs is *Kraken* [54], which has recently been updated to *Kraken2* [55]. It builds its database from NCBI RefSeq genomes [45, 46] and  by splitting those into k-mers. These k-mers are then mapped on the nodes of a phylogenetic tree: the more widely they are shared by multiple organisms the deepest they are mapped on the tree. The unique k-mers, seen only once in the whole database, are mapped on terminal nodes [54]. *Kraken* classifies genomic regions, here genomes, by cutting them into k-mers of the same length as those of the database, and mapping the k-mers on the tree [54]. The mapping path forms a subtree for each sequence, which allows *Kraken* to compute a sequence-specific taxonomic label (i.e. a LCA) [54]. *Kraken* has been designed as a read classifier but it has also been used to remove contaminants from assembled genomes after cutting genomes into pseudo-reads [56]. *Kraken2* can work on Bacteria/Archaea, human, fungi, plant, and viral genomic data at the same time [55]. *CLARK* uses a similar approach to classify sequences, with the exception that only signature k-mers, unique to a given taxon, are considered [57]. As there is no need to map shared k-mers on a phylogenetic tree, *CLARK* does not use it and instead classifies sequences at the genus and species level, with a sensitivity superior to *Kraken* [57]. Unlike *Kraken2*, *CLARK* works only on prokaryotic genomes (Fig. 2). This strategy of using only unique k-mers has also been implemented in *Kraken-uniq*,

but with the advantage to work on multiple taxonomic ranks [58]. To increase sensitivity, by matching more k-mers during classification, *Kraken2* offers the possibility to mask positions in k-mers during database construction [55]. Starting at the end of the k-mer and going back to the beginning, positions are masked in alternance until a specified number, 7 by default, is reached [55]. A recent tool, *CONSULT*, offers the best sensitivity among long k-mer detection tools by using the Hamming distance instead of exact matching [59]. *CONSULT* is currently restricted to prokaryotes and does not output the taxonomy of the sequences like *Kraken* or *CLARK* do [59] but it represents an interesting alternative, notably for rare genomes.

Genome-wide approaches also contain three tools (*BASTA*, *Physeter*, *GUNC*) that use *BLAST* [27] or *DIAMOND blast* [28] to perform gapped alignments against a reference database.

The first two programs (*BASTA*, *Physeter*) classify sequences using LCA labels, an approach again inspired by *MEGAN* [37]. The sequences from the genome under study are BLASTed against a database containing taxonomically labelled sequences. With *Physeter*, the query sequences can be cut into shorter "pseudo-reads" to increase sensitivity [3, 22]. The alignments are parsed to filter out database hits based on identity percentage, value, and/or bit-score thresholds [22, 60]. For each query sequence, the accumulated hits and their associated taxonomy are used to compute an LCA [60]. The main difference between *BASTA* and *Physeter* is the possibility to perform a leave-one-out analysis with *Physeter*, so as to reduce the impact of a potential contamination of the reference database [3]. To this end, the database is split into 10 parts and the LCA inference is run 10 times on 90% of the database. These two programs can work on prokaryotic and eukaryotic genomes at the same time, both in nucleotides and proteins [3, 60]. Finally, *Physeter* is one of the three programs of this review to be able to use the GTDB Taxonomy in addition to the commoner NCBI Taxonomy (Fig. 2).

The last program of this review (*GUNC*) uses the taxonomic homogeneity of contigs to infer contamination [1]. *GUNC* relies on a curated microbial database derived from the representative species of the proGenome2 database [61]. To infer the taxonomy of genes along contigs, *GUNC* retains only top *BLAST* hits, without filtration since a downstream scoring is applied [1]. The authors of *GUNC* have indeed developed two scores to assess the robustness of the contamination estimation, which have to be considered altogether when reading the estimates [1]. The first one, the Clade Separation Score (CSS), quantifies the degree of mixture of lineages [1]. A non-contaminated genome will have a CSS of 0, a genome with a different taxonomy for each contig (but homogeneous within the contigs) will have a CSS value of 1, and a genome with chimeric contigs will have intermediate CSS values [1]. The reference representation score (RRS) measures how confidently a given genome maps to the reference database, low RRS indicating novel (i.e. rare) lineages [1]. *GUNC* can work on prokaryotic genomes and is also able to use the GTDB Taxonomy [1] (Fig. 2).

## A detailed perspective on detection
### Negative vs positive filtering
Seven of the 18 tools presented in this review work by applying a "negative filter" on genome sequences. For this, they need an expected taxon, either specified by the user

(*BlobTools* [25, 26, 29], *Anvi'o* [30], *ProDeGE* [31], *PhylOligo* [33], *Forty-Two* [36, 42]) or determined by autodetection of the main organism (*Conterminator* [2], *BASTA* [60], *Physeter* [3]). Then, contaminants are detected when the input sequence taxonomy diverges from the expected taxon. The usage of "positive filtering", i.e. retaining only sequences with a specific taxonomy, can solely be achieved with four programs (*Kraken* [54, 55], *CLARK* [57], *BASTA* [60], *Physeter* [3]). All these tools indeed have the possibility to label taxonomically each individual sequence of a genome (or read for *Kraken* [54, 55] and *CLARK* [60]) and thus to apply a positive filter. In practice, a positive filter is useful when a researcher knows the expected taxon of the sequenced organism or after an overview of the taxonomy of the sequences found in the genome (based on a *Kraken*-like report, see Supplemental Note 1, available only for *Kraken* [54, 55], *CLARK* [60], and *Physeter* [3]). Such positive or negative filters are not included in algorithms based on gene markers (*ConFindR* [43], *CheckM* [48], *EukCC* [49], *BUSCO* [50], at the exception of *Forty-Two* [36, 42], or on the chimerical structure of the contigs (*GUNC* [1]) since these do not infer the taxonomy of the sequences.

### Correlation vs union

Formally, researchers can either use the intersection (i.e. corroboration) or the union of the results of multiple methods to assemble a list of contaminated genomes. The intersection can be used to assess that a given program is specific enough and has not produced false positives [3]. In this respect, database-free methods are useful, especially when only a few genomes from a key taxon are available. Nevertheless, the most frequent rationale for using multiple approaches is to increase the sensitivity and catch more contaminated genomes by considering the union of the methods. This is especially useful in large genomic projects where the loss of individual genomes is not too important.

### Comparison and benchmarking of algorithms

We tested and benchmarked a representative sample of six algorithms among the 18 presented in this review on artificial chimerical genomes containing both redundant and non-redundant as well as inter-domain contaminations (see Supplemental Note 1). We only selected algorithms relying on a reference database because those require no user interaction and are thus more convenient. First, we tested the tools based on gene markers. *CheckM* [48] and *EukCC* [49] are built on the same theoretical schema, the first one being specialized in prokaryotes [48] and the second in eukaryotes [49] *CheckM* and *EukCC* are the most widely used tools based on gene markers. We have selected *Forty-Two* [36, 42] because it is the only gene-marker-based software package able to perform inter-domain detection. Among genome-wide tools, *Kraken2* [55] is based on long k-mer matching and was selected because of its importance in the domain (the *Kraken* suite has been maintained for the last 10 years [62]) and its ability to perform inter-domain detection. The two last tools tested are *Physeter* [3], both because of its inter-domain support and ability to minimize the adverse effects of a contaminated reference database [3], and GUNC [1], because it is the only tool exploiting the chimerical structure of contigs in prokaryotes [1]. The Supplemental Note 1 comparing these tools shows typical command lines and output for each of them. We also

provide Singularity [63] definition files to help researchers with the installation and testing of these tools (Supplemental Note 1).

When a contamination occurs, it can be complicated to determine the source and the importance (in quantity) of the contamination. In Supplemental Note 1, a bacterial chimerical genome was constructed by adding 11% of a Firmicutes genome to a Gammaproteobacteria genome, equally distributed among redundant and non-redundant contaminants (see Supplemental Note 1 for detailed methods). This chimerical genome was further concatenated to a fungal genome, so as to simulate the interdomain contamination of a eukaryotic genome, here contaminated by a contaminated bacterial genome. As expected, the two uncontaminated bacterial genomes (Gammaproteobacteria and Firmicutes) are well classified by all the tools, at the exception of *EukCC*, which was designed for eukaryotic contaminants in eukaryotic genomes (Supplemental Note 1). Regarding the chimerical bacterial genome, different levels of contaminants are reported by the remaining five tools (1.65% for *CheckM*, 10% for *GUNC*, 25% for *Physeter*, 10.5% for *Kraken2* and 48.8% for *Forty-Two*). Only *GUNC* and *Kraken2* are close to the correct proportion of contaminants (11%). The most surprising result is the low contamination level reported by *CheckM*, 1.65%, which would make the genome pass below the standard recommendation of 5% to consider a genome as contaminated [64]. *CheckM* is a program designed to detect redundant contamination [1, 48], which is represented by half of the Firmicutes sequences (5-6%) in our artificial genome, but the contamination detected by *CheckM* is nonetheless lower than expected here. This highlights the need for considering the union of multiple methods to catch contaminants, as recently suggested [1, 3, 22]. Moreover, the different amounts of contaminants estimated on this artificial genome demonstrate that when using multiple tools to assess the contamination level, it is very difficult to compute meaningful correlations, a limitation that we have already explained in the past (see [3, 22]). The results for the reference fungal genome are more complicated to interpret. Inter-domain detection tools (*Forty-Two*, *Kraken*, *Physeter*) and *EukCC*, designed for eukaryotes, logically do not detect (too many) contaminants, but the programs calibrated on prokaryotes (*CheckM* and *GUNC*) identify this genome as a bacterium, *CheckM* inferring a low level of completeness and a high proportion of contamination, whereas *GUNC* reports a low level of representation in its database (Supplemental Note 1). Altogether, the values provided by *CheckM* and *GUNC* confirm that their algorithms are not suitable for this genome, since they were designed for prokaryotes. Inter-domain contamination (a fungus and a chimeric bacterium in the same file) can produce the same difficulties of interpretation. Hence, *CheckM* and *EukCC* both report a high contamination level (122.41% for *CheckM*, 14% for *EukCC*). *CheckM* classifies the genome as a bacterium, as with the purely eukaryotic genome, and finds a high proportion of contaminants, whereas *EukCC* classifies the genome as a eukaryote and finds less contaminants (Supplemental Note 1). These results are surprising as *CheckM* should not detect eukaryotic contaminants [48] while EukCC is not supposed to detect bacterial contaminants [1]. The false-positive detection is caused by similarities between prokaryotes and eukaryotes in the set of gene markers used (e.g. ribosomal proteins whether universal or Alphaproteobacterial, due to eukaryotic mitochondria). This shortcoming can lead to interpretation errors when analyzing

complex samples. *GUNC* finds few contaminants but also reports a weak representation score in its database and is thus less subject to drawing incorrect conclusions. The three inter-domain detection tools (*Forty-Two*, *Kraken*, *Physeter*) correctly identify the main organism (Opistokontha) and the two bacteria (Gammaproteobacteria and Firmicutes). These software packages are also able to provide a taxonomic analysis of the contaminants (Supplemental Note 1). Among the three tools, *Kraken2* is the only one reporting an accurate proportion of each organism after a normalization of sequence length (Supplemental Note 1). Rachtman et al. [59] have recently demonstrated that *CONSULT* performs better than *Kraken2* on rare genomes [59], by using the Hamming distance instead of exact matching of long k-mer. While *CONSULT* does not currently output the taxonomy and is restricted to prokaryotes, it might be a promising alternative for the years to come.

## Futures challenges

Improvement of the detection of inter-domain and sub-species contamination, as well as better detection of non-redundant contamination in future algorithms, would certainly broaden researchers' choice. However, other challenges remain complicated to address, notably the presence of taxonomic errors and rare genomes in databases, contamination of reference databases or the distinction between contamination and HGT events.

### Taxonomic errors and rare genomes

Genomes are downloaded from public repositories, often based on their declared taxonomy. Researchers interested in a specific taxonomic group can, in this way, easily obtain hundreds of genomes for comparative genomic or phylogenomic studies. Lupo et al. [3] demonstrated that mis-affiliated genomes are present in NCBI RefSeq. These genomes may not be contaminated sensu stricto, but the main organism is not the expected one. The inclusion of organisms from unwanted taxa in a study can obviously lead to artifactual results [16, 17]. However, these cases are not detected by the majority of algorithms, as the genome is not chimeric. In principle, *Physeter* is the only tool able to detect mis-affiliated genomes by automatically downloading the taxonomy associated to the genomes from NCBI or GTDB servers and comparing the declared organism with the main detected organism [3]. In practice, this approach only works partially because it is difficult to distinguish taxonomic errors from two other cases: (1) when the genome data is so heavily contaminated that the expected taxon becomes very scarce and (2) when dealing with rare genomes with no close representative in reference databases [3].

### Contamination of reference databases

Fourteen of the 18 programs presented in this review are associated to a reference database constructed from public genomes (Fig. 2). The contamination of a fraction of the genomes in these public repositories (Fig. 1) is an issue that has been demonstrated in the recent literature [3, 12, 22, 48, 65]. These algorithms are thus subject to false positives or negatives if they rely on wrongly classified sequences. This danger of incorrect inference (whether over- or under-detection) is especially true for tools using the

taxonomy of their reference database (*Forty-Two, Conterminator, Kraken, CLARK, CONSULT, BASTA, Physeter, GUNC*). Indeed, algorithms relying on an extra number of gene markers to infer contaminations (*ConFindR, CheckM, EukCC, BUSCO*) are less subject to incorrect conclusions due to a contamination of their database because they do not infer taxonomy. *Physeter* offers the possibility of using a leave-one-out approach when inferring contamination, which minimizes the adverse effects of individual contaminated genomes in the reference database [3]. Nevertheless, this approach is only successful if the database is rich enough to maintain genome diversity during the leave-one-out step. Rare taxonomic groups, typically only represented by a few genomes, might suffer from this strategy. Furthermore, if a taxonomic group is represented by numerous contaminated genomes in public repositories, the leave-one-out approach might not be enough to discard all of them at once. Therefore it is important to keep working on the curation of public databases, with the aim of making these sufficiently diversified from a taxonomic point of view but with a minimal amount of contamination, in order to maintain the efficacity of the detection algorithms in the future.

### Horizontal gene transfer and contamination

HGT is a natural cause of genetic material exchange between organisms. The rate of HGT in prokaryotes is expected to be high, since the majority of bacterial genes have been transferred at least once in the past [66, 67]. HGT has also been reported in gut-inhabiting microorganisms [68, 69], suggesting that such events also affect metagenomic samples. HGT is not limited to bacteria and can involve eukaryotes too. For example, numerous cases of gene exchange between bacteria and fungi have been reported [70–75]. Two different problems can be distinguished when considering HGT and contaminations. First, the detection of HGT is complicated by the background noise created by the contaminants. Studies have reported that contaminants can be mistaken for HGT candidates in tardigrade [6–8], rotifers [76], human [77] and, recently, in arthropods [78]. Secondly, in the opposite way, the detection of contaminants can be complicated by "genuine" HGT. Programs for contamination detection are thus prone to errors in this case, especially those that rely on reference databases for genomic comparison. The only tool that has been tested against HGT is *GUNC* [1]. Authors report less than 10% of false positives due to HGT events following the analysis of an HGT-enriched dataset derived from proGenomes2 (Khedkar et al., unpublished). Until now, genomic contamination and HGT have been considered as exclusive scenarios, a foreign sequence being either a contamination or the result of an HGT. Yet, a genomic region can actually undergo both types of events, depending on the considered taxon, which thus represents a challenge to be addressed in future algorithms.

### Other types of contamination

This review focused on the tools designed for the detection of foreign prokaryotic or eukaryotic sequences in assembled genomes. It is important to mention that the programs presented here are not able to detect all type of sequences. Indeed no study has been conducted with these tools on mobile elements in prokaryotes, such as plasmids, which are also subject to contaminating genomic data [79]. Furthermore, viral data have not been considered in this review.

## Supplementary Information

---

**Additional file 1: Supplemental Notes.** Comparison and benchmarking of six contamination detection tools. The note compares and benchmarks six of the 18 tools discussed in the article: *CheckM*, *EukCC*, *Forty-Two*, *GUNC*, *Physeter* and *Kraken2*. These programs are tested on artificial chimerical genomes containing both redundant and non-redundant as well as inter-domain contaminations. Typical command lines and output for each of these six tools are also provided.

**Additional file 2.** Review history.

---

### Acknowledgements
The authors thank Rosa Gago for her help with the graphical design of the figures.
We are grateful to the two anonymous reviewers for their insightful comments.

### Peer review information
Kevin Pang was the primary editor of this article and manage its editorial process and peer review in collaboration with the rest of the editorial team.

### Review history
The review history is available as Additional file 2.

### Authors' contributions
LC and DB conceived the study and wrote the manuscript. LC drafted the figures. The authors read and approved the final manuscript.

### Funding
This work was supported by a research grant (no. B2/191/P2/BCCM GEN-ERA) funded by the Belgian Science Policy Office (BELSPO). Computational resources were provided by the Consortium des Équipements de Calcul Intensif (CÉCI) funded by the F.R.S.-FNRS (2.5020.11), and through two research grants to DB: B2/191/P2/BCCM GEN-ERA (Belgian Science Policy Office - BELSPO) and CDR J.0008.20 (F.R.S.-FNRS).

### Availability of data and materials
Not applicable.

## Declarations

### Ethics approval and consent to participate
Not applicable.

### Competing interests
The authors declare that there are no conflicts of interest.

### Author details
[1]BCCM/IHEM, Mycology and Aerobiology, Sciensano, Bruxelles, Belgium. [2]InBioS–PhytoSYSTEMS, Eukaryotic Phylogenomics, University of Liège, Liège, Belgium.

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

## 