## [**Additional file 2.** Review history. · Genome Biology]

Review History

First round of review

Reviewer 1

I read the manuscript with much anticipation because the topic is important, and the authors have noticed a real gap; that is, there are many tools for dealing with contamination but little clarity in terms of their differences and strength. The paper is well written and helps a reader not greatly familiar with these tools (like myself) to gain some level of understanding. There is much value in having a review like this published.

High-level criticism: However, for all my enthusiasm, I came out of reading the paper with a sense that I did not learn as much as I had hoped. I give some more specific comments below. However, before doing that, I wanted to share my interpretation that the review, while quite thorough, lacks discussion of concepts and big picture debates, challenges, pitfalls, things the reader should be worried about, etc. It is excellent to go through all the algorithms, but I would have liked to see more general issues and the taxonomy of issues to consider. I understand that this comment may be hard to apply. So, here are more specific questions.

Detailed points to address:

1. Detection of contamination can happen at different stages: that is, on a read-by-read basis, on scaffolds, individual genes, etc. In particular, there is a big difference between looking for contaminants pre and post assembly. I think this distinction is not clear. I believe tools like Kraken would be useful mostly prior to assembly (i.e., they are run on the reads), while others like CheckM are post-assembly. Incidentally, I believe authors have emphasized this point in their prior work published in PeerJ.
2. Detection of contaminants can happen by 1) checking a sequence against what is expected to be there (call it a positive filter) or 2) checking a sequence against possible contaminants that should not be there but may be there (call it a negative filter). I think it would be helpful to make this clear. Also, I believe most of the methodology described here is for positive filters, though some of the tools, like Kraken and Clark, can be used for negative filters. For example, for a sample supposed to be a dog, you can look for bacterial reads in the set of reads.
3. There is a potential problem with checking reads or perhaps even scaffolds against a database of other sequences from the (supposedly) the same species: namely, the references themselves may be contaminated. Imagine you have *Drosophila*, and you want to find contaminant reads (or contigs) in that *Drosophila* genome. If you search your sequences against a library of other *Drosophila* genomes, some of which are contaminants with the same species (say, a common bacteria), then you may miss the contaminant. A reader would benefit from understanding this difficulty. I think the authors touch on this in passing, but a more detailed discussion would be good.
4. While authors do mention eukaryotes, there seems to be a focus on methods that address bacteria. Perhaps there is a lack of methods for Eukaryotes? If so, it would be good to mention this. If there are many methods for eukaryotes but not covered here, the focus should be clarified to the reader.
5. It would be helpful to have some figures (even if in supplement) that show example outputs from these tools (even if not every one of them). I often found myself trying to imagine what the output would look like based on the given description. Being able to see some of the outputs would be a great addition. It would be far more useful than Figure 1, which is not that useful in my judgment.

6. I would have liked to see or be directed to actual benchmarking results that compare these methods. A lot of the comparisons felt more like casual observations rather than claims backed by data. As a reader, I would appreciate the comparisons more if they were rooted in benchmarking results comparing these methods and if those results were clearly pointed out.

7. This paper is quite comprehensive as a review paper, and the authors have captured many of the relevant citations. However, some relevant citations are inevitably missed, which I would like to bring to your attention.

* Interesting examples of contamination found in the literature:

- Artamonova, Irena I., and Arcady R. Mushegian. "Genome sequence analysis indicates that the model eukaryote *Nematostella vectensis* harbors bacterial consorts." *Applied and environmental microbiology* 79.22 (2013): 6868-6873.

- Salzberg, Steven L., et al. "Serendipitous discovery of *Wolbachia* genomes in multiple *Drosophila* species." *Genome biology* 6.3 (2005): 1-8.

- Tsoktouridis, Georgios, George Tsiamis, Nikolaos Koutinas, and Sinclair Mantell. "Molecular Detection of Bacteria in Plant Tissues, Using Universal 16S Ribosomal DNA Degenerated Primers." *Biotechnology, Biotechnological Equipment* 28, no. 4 (2014): 583-91

- Wilson, Christopher G, Reuben W Nowell, and Timothy G Barraclough. "Cross-Contamination Explains 'Inter and Intraspecific Horizontal Genetic Transfers' between Asexual Bdelloid Rotifers." *Current Biology : CB* 28, no. 15 (August 2018): 2436--2444.e14.

* Studies on prevalence, theoretical modeling, and reviews:

- Francois, Clementine M., Faustine Durand, Emeric Figuet, and Nicolas Galtier. "Prevalence and Implications of Contamination in Public Genomic Resources: A Case Study of 43 Reference Arthropod Assemblies." *G3* 10, no. 2 (February 2020): 721-30.

- Eisenhofer, Raphael, et al. "Contamination in low microbial biomass microbiome studies: issues and recommendations." *Trends in microbiology* 27.2 (2019): 105-117.

- Rachtman, Eleonora, et al. "The impact of contaminants on the accuracy of genome skimming and the effectiveness of exclusion read filters." *Molecular ecology resources* 20.3 (2020): 649-661.

* Other methods for contamination removal:

- Rachtman, Eleonora, Vineet Bafna, and Siavash Mirarab. "CONSULT: Accurate Contamination Removal Using Locality-Sensitive Hashing." *NAR Genomics and Bioinformatics* 3, no. 3 (2021).

- Lee, Imchang, et al. "ContEst16S: an algorithm that identifies contaminated prokaryotic genomes using 16S RNA gene sequences." *International journal of systematic and evolutionary microbiology* 67.6 (2017): 2053-2057.

- Lu, Jennifer, and Steven L Salzberg. "Removing Contaminants from Databases of Draft Genomes." Edited by Fengzhu Sun. *PLOS Computational Biology* 14, no. 6 (2018): e1006277.

Minor

Line numbers given.

55 - 66: Can the distinction between the two be defined more precisely? I think I understand the concepts, but sentences here were a bit hard to parse. A more precise definition would help. The Figure was of no help in distinguishing the two for me.

171: sensibility? Perhaps you mean sensitivity?

214: This sentence is not accurate. Kraken does allow inexact matching (through masking), and also, it should not be called an alignment method, in my opinion.

Reviewer 2

This is a well written review focusing on the important issue of genome contamination. The authors summary of the different contamination detection algorithms will be of help to the broader microbial genomics community.

I have a few minor comments:

* Line 27: Consider adding Reference "Large-scale contamination of microbial isolate genomes by Illumina PhiX control. Pubmed ID: 26203331"

* Line 52: Consider adding Reference "Contamination Issue in Viral Metagenomics: Problems, Solutions, and Clinical Perspectives Pubmed ID: 34745046"

* Lines 163 - 172: Consider including reference "Within-species contamination of bacterial whole-genome sequence data has a greater influence on clustering analyses than between-species contamination - PMID 31849328"

* Lines 523 & 524 : Please expand the figure legends to be a little more descriptive, especially Figure 2.

* I think the authors can add a few sentences and reference for Plasmid DNA contamination (Example PMID: 30733546 Plasmid DNA contaminant in molecular reagents)

* I have used ContEst16S (PMID: 28639931) in the past and found it to be quite useful. It may be worth taking a look at the paper and include it in the current review

Figure 1: I feel this figure can be modified slightly to group the different sources of contaminants. This can be done either in the image or detailed in figure legend. Also please add a circle for NCBI SRA at the bottom, next to GenBank and RefSeq.

Reviewers' reports

> We thank the two reviewers for their detailed proofreading. The quality and interest of our review has been much improved by their suggestions.

Reviewer #1

I read the manuscript with much anticipation because the topic is important, and the authors have noticed a real gap; that is, there are many tools for dealing with contamination but little clarity in terms of their differences and strength. The paper is well written and helps a reader not greatly familiar with these tools (like myself) to gain some level of understanding. There is much value in having a review like this published.

High-level criticism: However, for all my enthusiasm, I came out of reading the paper with a sense that I did not learn as much as I had hoped. I give some more specific comments below. However, before doing that, I wanted to share my interpretation that the review, while quite thorough, lacks discussion of concepts and big picture debates, challenges, pitfalls, things the reader should be worried about, etc. It is excellent to go through all the algorithms, but I would have liked to see more general issues and the taxonomy of issues to consider. I understand that this comment may be hard to apply. So, here are more specific questions.

Detailed points to address

1. Detection of contamination can happen at different stages: that is, on a read-by-read basis, on contigs, scaffolds, individual genes, etc. In particular, there is a big difference between looking for contaminants pre and post assembly. I think this distinction is not clear. I believe tools like Kraken would be useful mostly prior to assembly (i.e., they are run on the reads), while others like CheckM are post-assembly. Incidentally, I believe authors have emphasized this point in their prior work published in PeerJ.

> The Reviewer is right. This distinction should be mentioned. Only three algorithms can be used before genome assembly (*Kraken*, *CLARK*, *CONSULT*) by filtering reads based on taxonomy. We have added this concept in the manuscript itself and in Figure 2.

[Lines 107-110 of the track-change document.]

2. Detection of contaminants can happen by 1) checking a sequence against what is expected to be there (call it a positive filter) or 2) checking a sequence against possible contaminants that should not be there but may be there (call it a negative filter). I think it would be helpful to make this clear. Also, I believe most of the methodology described here is for positive filters, though some of the tools, like Kraken and Clark, can be used for negative filters. For example, for a sample supposed to be a dog, you can look for bacterial reads in the set of reads.

> We have added a section in the manuscript to make the distinction between positive and negative filtering. In brief, most of the tools presented in the review apply a negative filter (filter what should not be present) and the algorithms using taxonomy for each individual sequence (such as Kraken) can apply a positive filtering. Software packages based on gene markers (ConFindR, CheckM, EukCC, BUSCO) track extra copies of unicopy markers and do not integrate positive or negative filters.

[Lines 442-483 of the track-change document.]

3. There is a potential problem with checking reads or perhaps even contains against a database of other sequences from the (supposedly) the same species: namely, the references themselves may be contaminated. Imagine you have drosophila, and you want to find contaminant reads (or contigs) in that drosophila genome. If you search your sequences against a library of other Drosophila genomes, some of which are contaminants with the same species (say, a common bacteria), then you may miss the contaminant. A reader would benefit from understanding this difficulty. I think the authors touch on this in passing, but a more detailed discussion would be good.

> Again, the Reviewer is right. Contamination of public databases is an important problem, especially for algorithms making use of a reference database (the majority of the tools). In this category, Physeter is the only package able to deal with database contamination by applying a leave-one-out approach. At first, to avoid highlighting our own tool in the review, we had not included a section on database contamination in the manuscript. However, as we believe that it is an important aspect for the future of the field, we have now added a subsection under the “Future challenges” section to clearly outline the interest and limitations of Physeter’s approach.

[Lines 615-638 of the track-change document.]

4. While authors do mention eukaryotes, there seems to be a focus on methods that address bacteria. Perhaps there is a lack of methods for Eukaryotes? If so, it would be good to mention this. If there are many methods for eukaryotes but not covered here, the focus should be clarified to the reader.

> Most of the methods are designed for prokaryotes. Nevertheless all the methods able to perform inter-domain detection (as indicated in Figure 2) can be used on Eukaryotes too, whereas EukCC and BUSCO have been specifically designed to work on Eukaryotes. We have now highlighted this in the legend of Figure 2 and added a definition of “inter-domain detection” before the overview of the algorithms.

[Lines 111-113 of the track-change document.]

[Lines 1062-1064 of the track-change document.]

5. It would be helpful to have some figures (even if in supplement) that show example outputs from these tools (even if not every one of them). I often found myself trying to imagine what the output would look like based on the given description. Being able to see some of the outputs would be a great addition. It would be far more useful than Figure 1, which is not that useful in my judgment.

> To address this useful suggestion, we have added a Supplemental Note that compares the output of the six most interesting tools, based on Figure 2. We chose tools that did not require user interaction, as those are the most often used by the community. In order to benchmark their efficiency, we have created artificial chimerical genomes, containing both redundant and non-redundant contamination, and covering both the bacterial and eukaryotic domain. In the Supplemental Note, we provide typical command lines and output of these software packages, as well as Singularity definition files and additional files required for reproducibility.

6. I would have liked to see or be directed to actual benchmarking results that compare these methods. A lot of the comparisons felt more like casual observations rather than claims backed by data. As a reader, I would appreciate the comparisons more if they were rooted in benchmarking results comparing these methods and if those results were clearly pointed out.

> Again, we thank the Reviewer for this stimulating criticism. This part of the manuscript has been replaced by a factual description of the behavior of the six algorithms used in the Supplemental Note. We believe that this section has become an interesting comparison of these popular tools, which illustrates their limitations and strengths.

[Lines 495-587 of the track-change document.]

7. This paper is quite comprehensive as a review paper, and the authors have captured many of the relevant citations. However, some relevant citations are inevitably missed, which I would like to bring to your attention.

> We thank the Reviewer for these interesting papers. We have added a part of them in the introduction to illustrate emblematic cases of contamination.

[Lines 42-47 of the track-change document.]

* Interesting examples of contamination found in the literature:

- Artamonova, Irena I., and Arcady R. Mushegian. "Genome sequence analysis indicates that the model eukaryote *Nematostella vectensis* harbors bacterial consort." *Applied and environmental microbiology* 79.22 (2013): 6868-6873.

> Added in the introduction.

[Line 43 of the track-change document.]

- Salzberg, Steven L., et al. "Serendipitous discovery of Wolbachia genomes in multiple Drosophila species." *Genome biology* 6.3 (2005): 1-8.

> Added in the introduction.

[Line 43 of the track-change document.]

- Tsoktouridis, Georgios, George Tsiamis, Nikolaos Koutinas, and Sinclair Mantell. "Molecular Detection of Bacteria in Plant Tissues, Using Universal 16S Ribosomal DNA Degenerated Primers." *Biotechnology, Biotechnological Equipment* 28, no. 4 (2014): 583-91

> Added in the introduction.

[Line 57 of the track-change document.]

- Wilson, Christopher G, Reuben W Nowell, and Timothy G Barraclough. "Cross-Contamination Explains 'Inter and Intraspecific Horizontal Genetic Transfers' between Asexual Bdelloid Rotifers." *Current Biology : CB* 28, no. 15 (August 2018): 2436--2444.e14.

> Added in the HGT sub-section.

[Line 649 of the track-change document.]

* Studies on prevalence, theoretical modeling, and reviews:

- Francois, Clementine M., Faustine Durand, Emeric Figuet, and Nicolas Galtier. "Prevalence and Implications of Contamination in Public Genomic Resources: A Case Study of 43 Reference Arthropod Assemblies." *G3* 10, no. 2 (February 2020): 721-30.

- Eisenhofer, Raphael, et al. "Contamination in low microbial biomass microbiome studies: issues and recommendations." *Trends in microbiology* 27.2 (2019): 105-117.

- Rachtman, Eleonora, et al. "The impact of contaminants on the accuracy of genome skimming and the effectiveness of exclusion read filters." *Molecular ecology resources* 20.3 (2020): 649-661.

> The two first papers have been added in the HGT sub-section of the "Future challenges" section. The last paper has been added in the introduction.

[Lines 646-651 of the track-change document.]

[Lines 47-48 of the track-change document.]

* Other methods for contamination removal:

- Rachtman, Eleonora, Vineet Bafna, and Siavash Mirarab. "CONSULT: Accurate Contamination Removal Using Locality-Sensitive Hashing." *NAR Genomics and Bioinformatics* 3, no. 3 (2021).

> **CONSULT**, published during the initial writing of the manuscript, has been now added to the review as a long k-mer detection tool.

[Lines 392-396 of the track-change document.]

- Lee, Imchang, et al. "ContEst16S: an algorithm that identifies contaminated prokaryotic genomes using 16S RNA gene sequences." *International journal of systematic and evolutionary microbiology* 67.6 (2017): 2053-2057.

> **ConEst16S** has now been added in the gene marker section, along with SINA.

[Lines 214-215 of the track-change document.]

- Lu, Jennifer, and Steven L Salzberg. "Removing Contaminants from Databases of Draft Genomes." Edited by Fengzhu Sun. *PLOS Computational Biology* 14, no. 6 (2018): e1006277.

> This paper describes a protocol using Kraken. We now cite this paper as an example of the usage of Kraken for contamination detection.

[Lines 364-366 of the track-change document.]

Minor

55 - 66: Can the distinction between the two be defined more precisely? I think I understand the concepts, but sentences here were a bit hard to parse. A more precise definition would help. The Figure was of no help in distinguishing the two for me.

> **Legend of Figure 1** has been expanded to be more explicit about the two kinds of contamination.

[Lines 1043-1055 of the track-change document.]

171: sensibility? Perhaps you mean sensitivity?

> **Sensitivity** has been used to replace sensibility throughout the manuscript.

214: This sentence is not accurate. Kraken does allow inexact matching (through masking), and also, it should not be called an alignment method, in my opinion.

> The most recent paper on Kraken, describing the idea of Kraken (<https://www.frontiersin.org/articles/10.3389/fbinf.2021.808003/full>) uses the words "exact matching". We thus decided to keep them in our manuscript but we mention that Kraken2 can also mask positions to increase sensitivity.

[Lines 374-392 of the track-change document.]

Reviewer #2

This is a well written review focusing on the important issue of genome contamination. The authors summary of the different contamination detection algorithms will be of help to the broader microbial genomics community.

> We thank the Reviewer for this positive assessment of our work.

I have a few minor comments:

* Line 27: Consider adding Reference "Large-scale contamination of microbial isolate genomes by Illumina PhiX control. Pubmed ID: 26203331"

**> Added in the introduction
[Line 34 of the track-change document.]**

* Line 52: Consider adding Reference "Contamination Issue in Viral Metagenomics: Problems, Solutions, and Clinical Perspectives Pubmed ID: 34745046"

**> Added in the introduction.
[Line 71 of the track-change document.]**

* Lines 163 - 172: Consider including reference "Within-species contamination of bacterial whole-genome sequence data has a greater influence on clustering analyses than between-species contamination - PMID 31849328"

**> Added in the ConFindR section.
[Lines 254-257 of the track-change document.]**

* Lines 523 & 524 : Please expand the figure legends to be a little more descriptive, especially Figure 2.

**> The legend of Figure 2 has been expanded.
[Lines 1058-1071 of the track-change document.]**

* I think the authors can add a few sentences and reference for Plasmid DNA contamination (Example PMID: 30733546 Plasmid DNA contaminant in molecular reagents)

**> We have added a section to mention other types of contamination, such as from plasmid or viral sources.
[Lines 665-671 of the track-change document.]**

* I have used ConEst16S (PMID: 28639931) in the past and found it to be quite useful. It may be worth taking a look at the paper and include it in the current review

**> ConEst16S has now been added in the gene marker section, along with SINA.
[Lines 214-215 of the track-change document.]**

Figure 1: I feel this figure can be modified slightly to group the different sources of contaminants. This can be done either in the image or detailed in figure legend. Also please add a circle for NCBI SRA at the bottom, next to GenBank and RefSeq.

> We thank the Reviewer for this good idea. Figure 1 has been modified following this suggestion. Moreover, NCBI SRA has been added and the legend has been expanded.

Second round of review

Reviewer 1

The authors have fully addressed the comments I provided on the previous version. In cases where they did not, they provided acceptable justification. I thank the author for the detailed and clear response letter. The review is improved a lot and will be a great contribution to the field.